# Translational Lifestyle Medicine Approaches to Cardiovascular–Kidney–Metabolic Syndrome

**DOI:** 10.3390/healthcare14010051

**Published:** 2025-12-24

**Authors:** Zacharias Papadakis

**Affiliations:** Department of Health Sciences and Clinical Practice, College of Health Professions & Medical Sciences, Barry University, 11300 NE 2nd Ave, Miami Shores, FL 33161, USA; zpapadakis@barry.edu

**Keywords:** personalized medicine, pharmacotherapy and lifestyle interventions, genomic and nutraceutical interventions, equity and SDOH

## Abstract

**Highlights:**

**What are the main findings?**
Cardiovascular–Kidney–Metabolic (CKM) syndrome arises from a complex interplay among heart, kidney, and metabolic dysfunction, worsened by factors like inflammation, oxidative stress, and insulin resistance.Lifestyle medicine interventions—focusing on nutrition, physical activity, stress management, sleep hygiene, social support, and avoidance of risky substances—can offer significant benefits in managing CKM-related risk factors such as high blood pressure, unhealthy lipid profiles, and poor glucose control, when implemented consistently.

**What are the implications of the main findings?**
Despite their potential benefits, lifestyle medicine strategies remain underutilized in clinical practice; integrating them as a core component of CKM prevention and management may meaningfully reduce risk burden.Future work and clinical pathways should emphasize personalized, integrative implementation (including digital health supports and sex-specific considerations) to address research gaps, reduce disparities, and optimize outcomes across diverse populations.

**Abstract:**

Cardiovascular–Kidney–Metabolic (CKM) syndrome arises from interrelated cardiovascular, renal, and metabolic pathways that require coordinated therapeutic strategies. This narrative review synthesizes recent systematic reviews, meta-analyses, and original studies to evaluate the translational application of lifestyle medicine (LM) for CKM management. Evidence indicates that LM interventions targeting the six pillars of practice (nutrition, physical activity, stress management, sleep, social support, and avoidance of risky substances) can improve blood pressure, lipid profiles, glycemic control, and weight, with benefits that complement or at times rival pharmacotherapy. We outline opportunities at the LM–drug interface, including sodium-glucose cotransporter-2 inhibitors and nutrient-stimulated hormone agents such as GLP-1 and GIP, and highlight the need to test synergy and sequencing with LM. Persistent implementation barriers include prioritization of drug-centric care and limited protocolized delivery; the AHA 5A model and digital health tools, including wearables that enable real-time feedback, provide practical routes for integration. Marginalized populations carry a disproportionate burden of CKM, underscoring the need for equitable, culturally tailored approaches. Sex-specific gaps, particularly in post-menopausal lipid metabolism and insulin sensitivity, point to the promise of genomic and nutraceutical personalization. Future work should use preregistered, adequately powered multimodal trials to establish durable, scalable pathways for CKM care.

## 1. Introduction

There is a global shift towards unhealthy habits, which, combined with a sedentary lifestyle, has been linked to metabolic syndrome (MetS), cardiovascular disease (CVD), kidney dysfunction, and other metabolic disorders, which often lead to increased risks of morbidity and mortality [1,2,3,4,5]. This unhealthy lifestyle places extra burden to the network of cardiovascular, renal, and metabolic abnormalities, which recently was coined as cardiovascular–kidney–metabolic (CKM) syndrome [4,6]. Even modest exercise-test blood-pressure elevations foretell incident hypertension, sharpening CKM risk stratification [7]. Our previous work has explored the intricate connections within this axis from a network physiology perspective, highlighting the role of exercise training [8,9]. For instance, by examining postprandial network interactions following high-intensity exercise, we have demonstrated the feasibility of this approach in capturing complex physiological responses [9,10]. To complement this broad framework, a companion review presents a CKM stage-specific exercise prescription approach (Stages 0–4), with emphasis on modality selection and dosing priorities across disease progression [11].

Cardiovascular diseases such as coronary artery disease (CAD), stroke, heart failure (HF), arrhythmias, and hypertension (HTN) are all linked to damage of the heart and blood vessels due to imposed atherosclerosis and arteriosclerosis [12,13]. In the United States, irrespective of gender, CVD remains the leading cause of death, with females presenting heterogeneity in conventional risk factors compared to males [14,15,16]. Irrespectively of sex or other social determinants of health (SDOH), the early CVD risk factors detection and management has been associated with reduced morbidity and mortality and improved quality of life of the affected individuals [17,18,19,20].

Chronic kidney disease (CKD) is another health issue that contributes both to associated health-related costs and to the outcome of other diseases and conditions, such as HTN, diabetes, and obesity, which ultimately increases the CVD-related mortality rates [21,22,23,24,25,26]. The reason for such an impact is that CKD diagnosis, in early stages, is difficult, even though the urinary albumin to creatinine ratio (UACR) of 30 mg/g or higher is used as early diagnostic indicator of glomerular injury [21,23]. Literature supports the evidence that people with higher normal UACR have higher CVD complications and mortality risks, with women exhibiting stronger associations, followed by younger individuals and higher educational attainment [27]. Another interesting statistic is the “Hispanic paradox” that highlights the non-significant association among Hispanics between higher incidence of CVD risk factors and low mortality rates [21,28,29]. Early detection and targeted treatment strategies through lifestyle medicine (LM) approaches are crucial in preventing CKD progression and reducing related cardiovascular risks and mortality rates [25,26,30,31,32].

Metabolic-related CVD, resulting from metabolic abnormalities linked to high-fat and high-carbohydrate diets and conditions like obesity, diabetes, hypertension, and atherogenic dyslipidemia, is a significant global driver of morbidity and mortality [24,33]. Collectively, all the aforementioned metabolic-related CVD abnormalities are defined as MetS, which is linked to CKD in both bidirectional and additive manner that ultimately exacerbates CVD [4,34,35].

Prior to CKM syndrome recognition, the same concept was described as cardiorenal syndrome, failing to include the metabolic part of the involved convergence of the related systems/organs [4,6,36,37]. The benefit of using the current definition is that it captures the complexity and knowledge gaps in disease mechanisms and SDOH impact, in contrast to a disorder that represents the simultaneous cardiovascular and renal issues, but without accounting for the chronological reliance [38]. It is crucial for public health to address the CKM syndrome with LM interventions to show a premise [4,6,32,38,39,40,41].

The American Heart Association (AHA), recognizing the importance of addressing the CKM syndrome, the interconnected dysfunction, and the disproportionate impact on marginalized communities and the SDOH that contribute to premature morbidity, has through internal funding to tackle the problem [4,18,19,37,42]. Lifestyle medicine approaches seem a promising area for effectively managing CKM syndrome, but more research is needed to bridge the gap in such a non-pharmacological approach that can have a positive impact on the CKM components [4,6,43]. On top of that, since by 2050 the USA is projected to have an obesity crisis, now it is more urgent to implement comprehensive lifestyle interventions [5,44].

Therefore, LM may offer a promising yet underutilized framework for transforming disease management in terms of CKM [45,46,47,48,49]. In this contemporary narrative review, we synthesize evidence from recent randomized controlled trials and systematic reviews on LM and innovative pharmacological interventions as complementary strategies for managing CKM syndrome. To our knowledge, this is the first review to explicitly integrate the AHA’s CKM staging framework with the six pillars of LM. By linking mechanistic pathways (Figure 1) with representative clinical and translational studies across pillars (Table 1), we aim to provide a practical, CKM-focused LM framework to inform both individualized care and health system–level strategies. For readers seeking detailed stage-specific exercise dosing, the full framework is provided in a companion Cardiovascular Diabetology review [11].

## 2. Current Knowledge Gaps

### 2.1. Understanding CKM Syndrome

Worldwide, CKM syndrome poses a significant challenge to healthcare systems due to the complex interplay of CVD, CKM, and metabolic disorders [6,32,50]. The widespread prevalence of CKM syndrome is striking, with nearly 90% of U.S. adults meeting the diagnostic criteria. This distribution spans various stages: 12.5% at Stage 0, 16.7% at Stage 1, 40.0% at Stage 2, 22.9% at Stage 3, and 8.9% at Stage 4 [39]. Recognizing how CKM syndrome progresses with age is essential, as those over 65 are more likely to reach advanced stages. From a lifestyle medicine perspective, prioritizing regular physical activity, a balanced diet, quality sleep, and stress management can help preserve muscle mass, cardiovascular health, and cognitive function, ultimately slowing disease progression [51]. Disparities exist in CKM syndrome prevalence based on sex and race, with notable variations where Stage 2 is more prevalent among women and Stage 3 predominates among men [39]. Further racial disparities reveal that non-Hispanic White and Asian individuals are more commonly in Stage 0, while non-Hispanic Black individuals are disproportionately represented in Stages 2 and 4 [39,40].

Central role to the CKM pathophysiology is the interactions between excess or dysfunctional adipose tissue with induced inflammation, presence of insulin resistance and oxidative stress [4,35,52,53]. Such tissues release proinflammatory and prooxidative compounds, causing damage to the arteries, heart, and kidneys, while also contributing to systemic conditions such as liver dysfunction [54,55,56]. Moreover, visceral fat accumulation aggravates insulin resistance, disrupts glucose metabolism and intensifies the symptoms of metabolic syndrome [53,56]. The accumulation of ectopic fat in organs such as the heart and kidneys adds to the complexity of CKM syndrome, contributing to arrhythmias, myocardial dysfunction, and systemic hypertension. These factors, in turn, accelerate atherosclerosis, kidney damage, and other related health issues [4,32,35,57]. The presence of common risk factors further complicates the interplay between the heart and kidneys, underscoring the need for more in-depth research into the complex pathophysiology of CKM syndrome [6,55,57,58]. The diverse nature of CKM syndrome reflects variations in metabolic risk across different weight categories and demographic groups, highlighting the role of social determinants of health in driving disparities. Deeper exploration is needed to understand genetic predispositions and epigenetic influences that shape individual risk profiles and contribute to variations in CKM syndrome outcomes [4,6,20,32,34]. Alarmingly, looming federal funding cuts jeopardize precisely these precision-LM pipelines [59]. These interrelated hemodynamic, metabolic, inflammatory, and neurohormonal processes are schematically summarized in Figure 1, together with the six pillars of Lifestyle Medicine that can modulate them within the broader context of social and commercial determinants of health.

### 2.2. Translational Approaches

Due to the inherent complexity of the CKM syndrome, its effective management requires a multifaceted approach targeting its pathophysiology and its interconnected pathways, such as inflammation, oxidative stress, insulin resistance, hyperglycemia, dyslipidemia, the renin–angiotensin–aldosterone system, and neurohormonal and vascular dysregulation [60]. Therefore, such multifaceted therapeutic approaches should integrate lifestyle modifications and pharmacological treatments to manage key components of the syndrome, such as blood pressure, lipid profiles, and glycemic control. Understanding the implicated lifestyle medicine mechanisms with their six pillars may be a novel therapeutic strategy that addresses the complexity of CKM. These lifestyle medicine approaches have been shown to be able to reduce inflammation and oxidative stress, elements that are key components of the syndrome. Moreover, personalized lifestyle medicine holds great promise for managing the CKM syndrome and improving the overall patient outcomes [6,49,61]. For example, acute aerobic exercise improves renal-health biomarkers even in moderate CKD, illustrating a clinically actionable ‘exercise first’ pathway [62,63,64]. However, the efficacy of such interventions depends heavily on precise dosing. A comprehensive evaluation of how moderate-intensity continuous exercise (MICE), high-intensity interval exercise (HIIE), and resistance training (RT) compare across different CKM stages is provided in our companion review [11], alongside evidence syntheses comparing interval vs. continuous training responses [65].

Current research often neglects the translational application of lifestyle interventions, especially their personalized application in managing CKM patients. Existing gaps include insufficient consideration of genetic and environmental factors, patient heterogeneity, absence of CKM—specific clinical trials, limited long-term outcomes data, inadequate integration strategies within healthcare systems, and minimal inclusion of SDOH [6,25,37,66,67]. Beyond these translational challenges, ensuring the replicability and methodological rigor of related research is crucial for building a reliable evidence base for lifestyle interventions [68,69,70,71]. Despite the supporting literature on the efficacy of lifestyle medicine in managing CKM syndrome, its application is still limited and inadequately integrated into standard medical practice, in contrast to pharmacological protocols and practices [4,57,72,73]. It is crucial to optimize the CKM syndrome management and improve patient outcomes. In an attempt to achieve this, a holistic and integrative lifestyle medicine approach is required. This approach should involve replicating findings for robust confirmation before widespread implementation, as evidenced by the diverse LM-related disciplines and subdisciplines [68,69,70,71,74,75]. Such an approach should focus on enhancing patient care, emphasizing interdisciplinary collaboration among the healthcare practitioners so that they will ultimately integrate these approaches into their daily practice. To illustrate how Lifestyle Medicine is already being tested in CKM-relevant populations, Table 1 presents representative interventions across different pillars, their primary mechanistic targets, key outcomes, and practical translational insights (including limitations) that can guide clinical implementation.

**Table 1 healthcare-14-00051-t001:** Translational Lifestyle Medicine evidence relevant to cardiovascular–kidney–metabolic syndrome.

Intervention Pillar	Study	Target Mechanisms Relevant to CKM Syndrome	Key Findings Relevant to CKM Outcomes	Translational Insight/Limitation
**Multimodal intensive lifestyle intervention (weight, diet, physical activity, behavioral counseling)**	Intensive primary-care lifestyle intervention in underserved adults with obesity [45] and multicenter RCT in adults with type 2 diabetes [76].	Reduction in overall adiposity (and central/visceral depots in imaging substudies of intensive lifestyle programs); improved insulin sensitivity and glycemic control; favorable changes in HDL-C and other cardiometabolic risk factors; modest effects on blood pressure.	In the pragmatic PROPEL trial, a 24-month intensive lifestyle intervention delivered by health coaches in primary care produced greater weight loss and clinically meaningful improvements in HDL-C and metabolic syndrome severity compared with usual care in a predominantly low-income, racially diverse population [45]. In Look AHEAD, intensive lifestyle intervention led to large, sustained weight loss and improvements in glycemic control and CV risk factors but did not significantly reduce major adverse cardiovascular events in the overall sample [76].	Demonstrates that guideline-concordant intensive lifestyle programs can be implemented in real-world primary care and improve CKM risk factors, particularly in underserved populations [45]. The absence of overall event reduction in Look AHEAD highlights the need to integrate lifestyle with optimal pharmacotherapy and to identify subgroups who may derive the greatest CKM event reduction [76].
**Physical activity and structured exercise**	Evidence-based exercise prescription across 26 chronic diseases [77]; exercise recommendations and vascular/autonomic responses in CKD, including acute exercise trials in moderate CKD [63,78,79,80].	Increased cardiorespiratory fitness; improved endothelial function and arterial stiffness; favorable shifts in autonomic balance; reductions in blood pressure, insulin resistance, visceral adiposity, and low-grade inflammation; in CKD, acute improvements in FMD and autonomic recovery without evidence of acute renal harm.	Across cardiometabolic conditions, 150–300 min/week of moderate-to-vigorous physical activity is associated with lower incidence of type 2 diabetes, improved blood pressure and lipid profiles, and reduced cardiovascular events in a clear dose–response pattern [77]. In non-dialysis CKD, supervised aerobic and resistance training is safe, improves exercise capacity and quality of life, and can modestly improve blood pressure and renal risk markers, though hard CKD and CV endpoints remain less well studied [78,79]. Acute crossover trials in moderate CKD show that a single bout of high-intensity interval or steady-state exercise augments flow-mediated dilation and improves cardiac autonomic recovery without signs of acute renal injury [63,80].	Supports formal exercise prescription as a core CKM therapy, including for CKD, consistent with “exercise is medicine” frameworks [77]. Underlines the importance of embedding brief PA counseling, referral pathways, and monitoring of aerobic and resistance training targets within routine CKM care, tailored to comorbidities and functional capacity [78,79]. CKD-specific acute-exercise studies suggest a therapeutic window in which vascular and autonomic benefits can be obtained without immediate renal hemodynamic compromise, but larger and longer-term trials are needed to confirm effects on CKM outcomes [63,80]. For a detailed guide on stage-specific exercise modalities and dosing, see [11,65].
**Nutrition—Mediterranean and plant-forward lifestyle patterns**	MEDLIFE cross-sectional analysis in US career firefighters [81] and whole-country cohort in Spain [82].	High intake of minimally processed plant foods, extra-virgin olive oil, nuts, and legumes; lower intake of refined and ultra-processed foods; favorable fatty acid profile, combined with habitual physical activity, adequate rest, and social/convivial habits; collectively improves dyslipidemia, insulin resistance, oxidative stress, and vascular function.	Among 249 U.S. firefighters, higher adherence to the 26-item Mediterranean lifestyle (MEDLIFE) index was associated with markedly lower odds of metabolic syndrome and more favorable total and LDL-C (cholesterol) and total-to-HDL cholesterol ratios [81]. In a national Spanish cohort, higher MEDLIFE adherence was associated with lower prevalence of metabolic syndrome and significantly reduced all-cause and cardiovascular mortality over follow-up [82].	Supports a Mediterranean-type lifestyle, integrating diet, movement, rest, and social connection, as a pragmatic Lifestyle Medicine strategy for CKM prevention and risk reduction in occupational and general populations [81,82]. Provides rationale for workplace and community interventions and for using tools such as MEDLIFE to operationalize lifestyle assessment in CKM clinical practice.
**Sleep–exercise–postprandial cardiometabolic control**	Experimental one-week sleep restriction in healthy adults [83,84] and acute partial sleep deprivation plus high-intensity exercise with high-fat feeding [9].	Sleep restriction and circadian disruption impair insulin signaling, increase sympathetic activity and blood pressure, alter heart-rate variability, and worsen postprandial metabolic responses. Under conditions of acute sleep loss and high-fat intake, high-intensity exercise may modify or blunt expected cardioprotective patterns in brain–heart–metabolic coupling.	One week of restricting sleep to ~5–6 h/night in healthy adults reduced insulin sensitivity by ~20–25%, impaired glucose tolerance, and increased evening blood pressure changes consistent with a pre-diabetic phenotype independent of weight [83,84]. In a within-subject crossover study, acute partial sleep deprivation followed by morning high-intensity interval exercise and a high-fat breakfast altered network interactions between heart-rate variability and LDL cholesterol, suggesting modification or blunting of usual cardioprotective exercise patterns under sleep-deprived conditions [9].	Elevates sleep and circadian health to a core Lifestyle Medicine pillar in CKM, indicating that exercise prescriptions and meal timing may need modification in sleep-restricted individuals [83,84]. The findings of Papadakis et al. [9] imply that high-intensity exercise performed after acute sleep loss and concurrent high-fat intake may not yield the full expected cardiometabolic benefit, reinforcing the need for coordinated interventions across sleep, exercise, and nutrition domains.
**Positive social connection, loneliness, and lifestyle engagement**	Objective activity and social isolation in older adults [85] and outcome-wide longitudinal analysis of loneliness and social isolation [86].	Social isolation and loneliness reduce daily physical activity and increase sedentary time; disrupt neuroendocrine and inflammatory pathways; worsen depressive and anxiety symptoms; indirectly amplify CKM risk and undermine adherence to lifestyle and medical therapies.	In 267 older adults from the English Longitudinal Study of Ageing, greater social isolation (but not loneliness) was associated with lower 24 h activity counts, more sedentary time, and less light and moderate-to-vigorous physical activity objectively measured by accelerometry [85]. In a large US cohort of older adults, longitudinal changes in loneliness and social isolation were independently associated with multiple adverse physical, behavioral, and psychological outcomes; social isolation more strongly predicted mortality risk, whereas loneliness more strongly predicted psychological outcomes [86].	Positions positive social connection and mental health support as essential components of Lifestyle Medicine in CKM, supporting routine screening for isolation and loneliness and incorporation of group-based LM programs and community linkage [85,86]. Reinforces the need for team-based care that integrates behavioral health, social work, and community partnerships to sustain lifestyle change and CKM risk reduction.
**Avoidance of harmful substance use (tobacco and high-risk alcohol)**	Smoking and CKD/CVD risk in large cohorts and meta-analyses [87,88], and prospective cohorts of alcohol consumption, all-cause mortality, CVD, and CKD [89,90], and large genetic–epidemiologic analyses of alcohol intake and CVD risk [91].	Cigarette smoking promotes oxidative stress, endothelial dysfunction, and activation of the renin–angiotensin–aldosterone system; accelerates atherosclerosis; and contributes to glomerular injury and CKD progression. Smokeless tobacco products may also worsen CKM risk factors. High-risk alcohol use increases blood pressure, atrial fibrillation, cardiomyopathy, and overall cardiovascular mortality; relationships between low-to-moderate alcohol use and CKD/CVD risk are complex and may vary by CKM stage.	Meta-analyses and community-based cohorts show that current smoking is independently associated with a higher risk of incident CKD and faster decline in renal function, with risk increasing with cumulative exposure [87,88]. Studies of smoking timing suggest that lighting the first cigarette soon after waking, especially in the context of a poor diet, may further amplify CKD risk [92]. Prospective cohorts of alcohol consumption report U-shaped or inverse associations with CKD risk but increased CVD and all-cause mortality at higher intakes, and recent cohort and genetic–epidemiologic analyses do not support recommending alcohol consumption for cardiometabolic benefit [89,90,91].	Underscores tobacco cessation as a non-negotiable Lifestyle Medicine priority across all CKM stages, given strong and consistent evidence that smoking increases CKD and CVD risk. Supports clear messaging that there is no safe level of tobacco use and that cessation should be aggressively supported with behavioral and pharmacologic tools. High-risk alcohol use should be systematically screened for and addressed as part of CKM management; contemporary evidence and recent analyses caution against recommending alcohol consumption for cardiometabolic benefit, particularly in patients with established CKM, and emphasize focusing on other lifestyle pillars instead [89,91].
**CKM framework and systems-level implementation of Lifestyle Medicine**	AHA Presidential Advisory on CKM health [4] and conceptual commentary on CKM syndrome and multidisciplinary care [93].	Integrated staging of CKM risk from optimal health to symptomatic disease; emphasis on interdependent pathways between obesity, dysglycemia, hypertension, chronic kidney disease, and heart failure, with lifestyle behaviors as upstream drivers; positioning Lifestyle Medicine as foundational therapy across CKM stages.	The CKM advisory proposes CKM stages, recommends systematic assessment of CKM risk factors, and explicitly elevates lifestyle interventions as foundational therapy at every stage of CKM risk, alongside pharmacologic risk-factor modification [4]. Claudel and Verma (2023) [93] argue that the CKM construct is an opportunity to build multidisciplinary, equity-focused care models that integrate Lifestyle Medicine, cardiology, nephrology, endocrinology, and primary care.	Provides a policy-relevant scaffold for embedding Lifestyle Medicine into CKM prevention and treatment, aligning clinic, community, and policy actions and incentivizing team-based care and quality metrics that reward lifestyle implementation [4,93]. This framing can help health systems and payers justify Lifestyle Medicine programs as core CKM infrastructure rather than optional adjuncts.

**Note.** CKD, chronic kidney disease; CKM, cardiovascular–kidney–metabolic; CV, cardiovascular; CVD, cardiovascular disease; FMD, flow-mediated dilation; HDL-C, high-density lipoprotein cholesterol; LDL-C, low-density lipoprotein cholesterol; PA, physical activity; RCT, randomized controlled trial; LM, Lifestyle medicine. Table 1 summarizes exemplar interventions across major pillars of Lifestyle Medicine, including nutrition, physical activity, sleep and circadian health, social connection, and avoidance of harmful substance use, together with multimodal and systems-level CKM framework approaches, highlighting the primary target mechanisms, key study findings, and translational insights and limitations.

### 2.3. Opportunities for Novel Therapeutic Strategies

The 21st century, with the great technological advancements, is in the best position to facilitate a comprehensive treatment framework-network strategy in managing CKM syndrome. Such strategies may enhance personalized health monitoring and serve both as educational and training tools among healthcare professionals and patients [94]. Now more than ever, individuals and patients are in a position to actively participate in their respective medical interventions through informed, continuous monitoring and real-time feedback. This feedback enables immediate adjustments to personalized lifestyle interventions based on dynamic data from wearable devices or health apps, significantly improving therapeutic adherence and CKM outcomes [43,49,95].

In the discussion of novel approaches to manage CKM syndrome through the lifestyle medicine perspective, the recent pharmacotherapies such as sodium-glucose cotransporter-2 (SGLT2) inhibitors and nutrient-stimulated hormones (NuSH) need to be mentioned as they do play a critical role in metabolic risk factors associated with CKM. These inhibitors have demonstrated potential in lowering blood glucose independently of insulin, while also offering possible renal protection and cardiovascular benefits. However, there is a lack of research exploring their potential synergistic or antagonistic interactions with the six pillars of the lifestyle medicine approach for managing CKM syndrome [4,46,61,64,96,97]. By promoting glucosuria, SGLT2 inhibitors help reduce visceral fat, a significant contributor to metabolic dysfunction, and diminish the risk of adverse cardiovascular outcomes. However, it is unknown whether this benefit is mediated or modulated by the effects of diet, sleep, or physical activity on visceral fat [30,35,46]. Similarly, agents such as glucose-dependent insulinotropic polypeptide (GIP) and glucagon-like peptide-1 (GLP-1) offer promising potential for reducing body weight and improving metabolic health. Their effectiveness is attributed to their ability to regulate insulin secretion in a glucose-dependent manner, suppress appetite, and promote weight loss, making them valuable for CKM management. However, there is uncertainty regarding how these agents interact with individual or combined components of the six pillars of lifestyle medicine [61,96,98,99,100,101].

Moreover, managing inflammation and oxidative stress through anti-inflammatory agents is crucial, as these factors drive insulin resistance and subsequent metabolic dysregulation. Therefore, along with these agents, lifestyle-targeted approaches have the potential to attenuate the progression of both CKD and CVD, potentially improving CKM outcomes by reducing oxidative stress and vascular dysfunction [4,35,43,53,58,102].

Dyslipidemia is a key component of CKM syndrome and is strongly linked to an increased risk of atherosclerotic cardiovascular disease (ASCVD), particularly in stages 3 and 4. The standard treatment approach often involves lipid-lowering medications such as statins and PCSK9 inhibitors, with combination therapy proving effective in significantly reducing low-density lipoprotein cholesterol (LDL-C) in high-risk patients, thereby supporting cardiovascular and kidney health. However, lifestyle medicine and nutraceuticals offer a side-effect-free alternative with the potential to achieve similar LDL-C reductions and, in severe cases, serve as a complementary therapeutic strategy for managing dyslipidemia [4,103,104,105,106].

Effective diabetes management is essential for CKM syndrome, as it directly impacts both CKD and CVD. In this context, SGLT2 inhibitors and GLP-1 receptor agonists provide metabolic and cardiovascular benefits by improving hyperglycemia and insulin resistance. These therapies effectively reduce excess adiposity and enhance insulin sensitivity, offering protective effects on both cardiovascular and kidney function. Incorporating a lifestyle medicine approach further strengthens this strategy by providing a holistic, multidisciplinary framework that integrates diabetes management, anti-inflammatory interventions, antioxidants, and lipid-lowering therapies. This comprehensive approach addresses the root causes of metabolic, renal, and cardiovascular dysfunction, ultimately improving CKM syndrome outcomes [13,38,43,100].

A new promising frontier in personalized healthcare involves the integration of genetic profiles with lifestyle medicine. As such, tailored lifestyle interventions based on diet and sleep [64] and under the individual’s genetic predisposition may have the potential to manage chronic diseases and subsequently improve CKM outcomes [67,107,108,109,110,111,112].

When exploring novel therapeutic strategies to combat CKM syndrome, it is crucial to consider patient adherence to lifestyle medicine approaches. Integrating these strategies into CKM management, similar to their implementation in cardiac rehabilitation programs for MetS, prediabetes, and weight management, could effectively address cardiovascular, renal, and metabolic risks associated with CKM. This holistic framework would support healthcare providers in facilitating long-term behavioral changes, ultimately improving patient outcomes [49,67,113,114,115,116,117,118].

Finally, SDOH must be incorporated into personalized intervention strategies. Models such as the Open-Source Wellness (OSW) demonstrate effectiveness in integrated, community-based interventions, improving health metrics and addressing societal health barriers [37]. Addressing socioeconomic disparities and ensuring access to nutritious food resources is essential for equitable implementation of lifestyle interventions, enhancing adherence and outcomes [37,119]. Future investigations should focus on identifying molecular pathways and physiological mechanisms through which lifestyle modifications impact CKM health, facilitating targeted interventions and enhancing patient response predictions [4,38].

### 2.4. Molecular Mechanisms of Lifestyle Interventions

To bridge the gap between clinical outcomes and pathophysiology, it is essential to delineate the molecular pathways through which lifestyle modifications exert their protective effects. Physical activity acts as a potent physiological regulator that upregulates mitochondrial biogenesis and antioxidant defense systems [60,120,121]. Mechanistically, aerobic exercise increases laminar shear stress, which activates the phosphoinositide 3-kinase (PI3K)/Akt pathway, leading to the phosphorylation of endothelial nitric oxide synthase (eNOS) and enhanced nitric oxide bioavailability [60,122]. Concurrently, skeletal muscle contraction stimulates the AMP-activated protein kinase (AMPK)–peroxisome proliferator-activated receptor γ coactivator 1α (PGC-1α) axis. This pathway is critical for driving energy metabolism, reducing systemic insulin resistance, and modulating inflammation via inhibition of the nuclear factor-κB (NF-κB) cascade [121,123,124]. These intracellular pathways also operate within broader inter-organ signaling networks; exercise-induced myokines/exerkines and muscle–kidney crosstalk are discussed as emerging mechanistic targets in CKM progression [11,65,125].

Nutritional interventions, particularly those emphasizing plant-based functional foods, target complementary molecular mechanisms. Polyphenols and flavonoids exhibit pleiotropic effects by activating the nuclear factor erythroid 2-related factor 2 (Nrf2) pathway, thereby upregulating endogenous antioxidant enzymes such as superoxide dismutase (SOD) and heme oxygenase-1 [124,126,127]. Beyond these broad dietary patterns, emerging preclinical work highlights the “exercise-mimetic” potential of specific nutrients. For example, caffeine has been shown in myotube and high-fat–diet animal models to activate AMPK–PGC-1α signaling, increase the expression and secretion of the myokine irisin/FNDC5, and promote the browning of white adipose tissue with associated improvements in adiposity and circulating lipid profiles [128]. Although these data are preclinical, they provide mechanistic proof-of-concept that targeted nutritional strategies may synergistically enhance the metabolic and thermogenic adaptations typically induced by physical activity. Collectively, these converging AMPK/PGC-1α-, Nrf2-, and eNOS-mediated responses provide a unified mechanistic basis through which Lifestyle Medicine interventions can modulate inflammation, oxidative stress, insulin resistance, and endothelial dysfunction in CKM syndrome [120,122].

## 3. Integrations of Lifestyle Medicine

The implementation of the six pillars of LM plays a crucial role in enhancing management strategies for CKM syndrome, predominantly focusing on nutrition and physical activity. Research supports the effectiveness of intensive structured lifestyle interventions in promoting weight loss and improving cardiometabolic health indicators more effectively than conventional care over extended periods, such as 24 months, highlighting potential long-term benefits in reducing cardiovascular disease risk factors [45].

An advisory from the AHA further emphasizes incorporating lifestyle counseling into routine clinical practice, aiming to improve cardiovascular health outcomes via the “5A Model” (i.e., Ask, Assess, Advise, Agree, and Assist) of behavior change. This model is designed to mitigate risks associated with cardiovascular diseases by encouraging patients to engage in healthier lifestyle habits. Healthcare providers are integral to this process, charged with the responsibility of engaging patients meaningfully in lifestyle counseling and supporting them through substantial lifestyle modifications that can enhance cardiovascular health [48].

Moreover, addressing the genetic and environmental factors and understanding patient heterogeneity are essential aspects that need more focus. Comprehensive trials tailored to CKM populations and integrating social determinants of health in lifestyle medicine can further narrow these gaps and improve the long-term success of lifestyle interventions. This holistic approach not only focuses on individualized strategies but also on creating sustainable health behavior changes that align with the broader public health guidelines and personal well-being goals of patients.

### 3.1. Nutrition

Western diets rich in fats and sugars negatively impact metabolic health, contributing to obesity, insulin resistance, and chronic conditions such as cardiovascular and kidney disorders. These diets elevate oxidative stress and inflammation, leading to metabolic dysfunction through increased reactive oxygen species that affect adipose and liver tissues. In contrast, polyphenol-rich diets like the Mediterranean, vegetarian, Asian, traditional Nordic, and DASH diets act as anti-inflammatory agents, counteracting the adverse effects of Western diets [49,126,129,130,131]. Optimal nutrition significantly influences blood glucose levels, insulin resistance, and type 2 diabetes mellitus management. Dietary composition and timing are critical; for instance, higher protein and fat intake at breakfast can improve HbA1c and glucose levels, while alternate-day fasting supports weight loss and insulin sensitivity. Incorporating whey protein has shown promise in reducing HbA1c and enhancing insulin sensitivity without altering fasting glucose levels [24,49,131,132].

Adhering to a Mediterranean diet, which emphasizes lower carbohydrate intake and is rich in plant proteins and healthy fats, effectively manages hyperglycemia and insulin resistance. A study on U.S. firefighters showed that adherence to the MEDLIFE index, reflecting a Mediterranean lifestyle, was linked to lower odds of metabolic syndrome and favorable cardiometabolic outcomes [81].

Precision personalized nutrition, encompassing nutrigenomics and nutrigenetics, is becoming fundamental in preventive and tailored healthcare strategies. While the genome and gut microbiome are primary focuses, epigenomic precision nutrition involving DNA methylation emerges as a novel biomarker for personalized interventions [42,111,133]. DNA methylation affects gene expression, influencing the development and progression of cardiovascular and metabolic diseases. For example, personalized anti-inflammatory nutritional interventions, including long-chain n-3 polyunsaturated fatty acids, can modulate adiponectin regulation and enhance dietary intervention effectiveness, particularly among individuals with adverse metabolic profiles [42,49,53,67,111,133,134,135].

Moreover, curcumin, a polyphenol in turmeric, offers anti-inflammatory and antioxidant benefits, especially in obesity management. Embracing a Mediterranean lifestyle, which incorporates a diet rich in vegetables, fruits, whole grains, and olive oil, correlates with reduced metabolic syndrome prevalence and improved cardiometabolic health [124,127,136].

The AHA recommends diets rich in vegetables, fruits, whole grains, and lean proteins while minimizing saturated fats, added sugars, and sodium to promote heart health. Similarly, the American Diabetes Association (ADA) emphasizes weight loss for insulin resistance management, advocating for calorie-restricted diets and increased fiber intake [24,49,81,137,138,139]. A multi-level approach to CKM syndrome management can optimize health outcomes, with personalized dietary recommendations considering age, sex, medical conditions, and genetic data to address individual needs effectively, thereby enhancing overall well-being [111,140].

While individual dietary choices are fundamental, they occur within a broader ecosystem shaped by the commercial determinants of health. Recent analyses, particularly the Lancet series on ultra-processed foods (UPFs) [141,142,143], highlight that the global rise in CKM syndrome is driven not merely by a failure of individual willpower, but by an industrial food environment that prioritizes profit over metabolic health. Engineered for hyper-palatability and durability, UPFs disrupt gut–brain signaling and satiety mechanisms, thereby driving excess calorie consumption and insulin resistance. The ubiquity of these products, coupled with aggressive marketing, creates a systemic barrier to the adoption of the whole-food, plant-forward diets recommended for CKM management [141,142,143]. Consequently, effective LM interventions must extend beyond individual counseling to advocate for policy-level changes, such as front-of-package labeling and marketing restrictions, that counter this obesogenic environment.

### 3.2. Physical Activity

Excess body weight and dysfunctional adiposity pose global public health concerns, linked to adverse health outcomes affecting various bodily systems. As of 2020, only 24.2% of adults met the physical activity guidelines, which recommend at least 150 min of moderate-intensity or 75 min of vigorous-intensity aerobic activity weekly, alongside muscle-strengthening exercises [44]. Emphasizing the significance of physical activity, substantial evidence supports its role in preventing and treating obesity and excess body weight [46,144,145]. The American College of Sports Medicine (ACSM) guidelines advocate for a minimum of 150 min per week of moderate-intensity activity to prevent weight gain and facilitate weight loss, resulting in improved weight management in a dose-responsive manner. While exercise without dietary restriction results in modest weight loss, it significantly impacts adipose tissue by reducing fat levels through aerobic and resistance exercises. Targeting visceral adiposity, physical activity reduces risks associated with metabolic diseases, highlighting the role of structured physical movement in treating and preventing excessive adiposity [8,46,144,146]. For example, the intensity of an exercise bout is a critical factor, influencing the magnitude of excess post-exercise oxygen consumption and, consequently, the attenuation of postprandial blood lipids, a key consideration for mitigating cardiometabolic risk in otherwise sedentary individuals [147].

Chronic inflammation associated with obesity exacerbates health risks, but regular physical activity reduces inflammatory markers, mitigating inflammation-related risks. This positioning of physical activity as a preventive measure against chronic inflammation and associated health complications is critical [55]. Appropriately combined aerobic and resistance exercises enhance insulin sensitivity and glucose metabolism, underscoring physical activity’s vital role in diabetes prevention and management. Improvements occur independently of weight loss, emphasizing exercise’s crucial role in metabolic control and reducing diabetes risks through effective blood glucose regulation [46,121,144,145].

Physical activity also significantly impacts oxidative stress, improving antioxidant defenses and reducing oxidative damage critical for preventing cellular harm and maintaining overall health, particularly relevant to obesity-induced oxidative stress. Excessive adiposity elevates the risk of CKD, but regular physical activity supports weight management and enhances kidney health by slowing disease progression, improving cardiovascular health, and reducing hypertension, a common CKD comorbidity [4,46,105,148]. Indeed, our investigations in moderate, non-dialysis CKD populations have shown that an acute aerobic exercise can improve vascular endothelial function, elicit favorable cardiac autonomic responses, and achieve this without adverse effects on biomarkers of renal health or filtration [62,63,80].

Metabolic syndrome symptoms are inherently linked to physical inactivity. Increasing physical activity levels effectively reduces risks associated with MetS by addressing obesity and insulin resistance while enhancing blood lipid profiles and glucose metabolism, crucial for effective MetS management [46,121,144,145]. Although seminal papers by authors such as Ndumele et al. (2023) [4] and Zoccali (2025) [38] emphasize the broad importance of physical activity in CKM management, they do not specifically address its targeted role.

Physical activity effectively influences CKM syndrome’s interrelated risk factors CKD, CVD, and metabolic dysfunction. This interconnectedness of systems and the impact of exercise align with insights from network physiology, as further elaborated in our prior research [8]. Increasing moderate-to-vigorous activity significantly reduces CVD risk, highlighting the importance of addressing excess and dysfunctional adiposity. Exercise reduces visceral adiposity and mitigates proinflammatory and prooxidative processes, contributing to the correction of metabolic dysfunction [4,8,46,149].

Physical activity is key in the primordial prevention of CKM syndrome, particularly at Stage 0, where optimal cardiovascular health is maintained without CKM risk factors. Regular activity helps sustain a healthy weight, blood pressure, glycemia, and lipid profiles, significantly preventing metabolic risks associated with CKM syndrome [4,121,145]. Additionally, promoting regular physical activity can enhance metabolic health, reduce the risk of diabetes, and mitigate CKM-exacerbating factors by improving glucose metabolism and insulin sensitivity [31,121,145,150].

A physically active lifestyle is strongly associated with reduced risks for both CVD and CKD [4,46,144]. With structured and varied physical activities being key components in weight management and cardiovascular rehabilitation, physical activity enhances life quality, exercise capacity, and reduces hospitalization instances. It attenuates myocardial oxidative stress and cardiac hypertrophy, while promoting beneficial metabolic adaptations and vascular adaptations, such as cardiac angiogenesis and endothelium-dependent vasodilation. These mechanisms, thus, collectively protect the myocardium and improve cardiovascular health [46,57,60,151]. Recent work has addressed this gap by proposing a CKM stage–specific exercise framework (Stages 0–4), detailing how modalities such as interval training and resistance training can be matched to stage-specific pathophysiology and clinical priorities [11].

### 3.3. Stress Management

Chronic stress significantly contributes to various health conditions, including cardiovascular disease and metabolic disorders like diabetes and obesity [24]. Meditation, mindfulness, and yoga are effective stress reducers, improving health outcomes by enhancing parasympathetic activity and lowering sympathetic responses that are often heightened during stress. Regular practice of mindfulness and yoga improves glycemic control, blood pressure, and body weight, making them valuable adjuncts in managing chronic health conditions [24,152,153,154]. This aligns with our recent findings linking mindfulness, mental toughness and self-compassion in student athletes [155].

An integrative approach that includes yoga, meditation, and mindfulness aids stress management and positively influences chronic disease biomarkers. This approach can reduce inflammation and oxidative stress, which are pivotal in the progression of diseases like CVD and diabetes [154] conditions [24,152,153,154]. Mindfulness-based stress reduction (MBSR) enhances both psychological and physical well-being by improving mood, reducing anxiety and depression, and consequently enhancing overall quality of life [153,154].

Incorporating yoga and meditation into daily routines provides a holistic strategy to combat the detrimental effects of chronic stress on cardiovascular and metabolic health [154]. This comprehensive stress management strategy aligns well with broader health goals, emphasizing personalized and integrative therapeutic interventions to mitigate the impact of stress on health.

### 3.4. Sleep Management

Adequate sleep duration and quality are essential for regulating hormones, promoting energy balance, and enhancing cognitive function. Sleep significantly impacts hormones involved in appetite regulation, such as leptin and ghrelin; insufficient sleep increases ghrelin and decreases leptin levels, elevating hunger and appetite, often resulting in weight gain [53,120,156,157,158]. Our laboratory demonstrated that combining partial sleep deprivation with high-intensity interval exercise amplifies short-term metabolic stress but preserves autonomic balance [159].

Short sleep duration disrupts leptin regulation, resulting in decreased leptin levels and increased ghrelin levels, which elevate hunger and are linked to weight gain and an increased risk of type 2 diabetes. Insufficient sleep negatively affects glucose regulation by impairing insulin sensitivity and increasing insulin resistance, contributing further to diabetes risk. Both short and long sleep durations are associated with poor glycemic control and can lead to hormonal imbalances such as elevated cortisol, exacerbating weight gain and metabolic disorders [157].

Sleep disruption impacts metabolic physiology by contributing to dysregulated eating behaviors and reducing energy expenditure, thereby affecting weight and cardiometabolic health [156,160,161]. Sleep also influences kidney function, with disturbances linked to the progression of CKD. Effective sleep management is crucial in preventing and managing CKD due to obesity-related poor sleep patterns [26,120,162].

The interplay between inadequate sleep, obesity, and kidney strain highlights the importance of adopting sleep hygiene practices to mitigate the risks of weight gain, obesity, and CKD, particularly when combined with exercise [8,138,159,163,164]. The interaction is complex; for instance, our work demonstrates that while acute partial sleep deprivation can impair postprandial endothelial function [148], high-intensity interval exercise performed in a sleep-deprived state can still elicit positive cardiac autonomic modulation and metabolic responses, highlighting exercise as a potential countermeasure to the acute cardiometabolic stress of poor sleep [159,164].

A short-term multi-component approach that targets diet, physical activity, and sleep in overweight and obese adults can aid in weight management and reduce cardiometabolic risks [161]. This integrated model underscores the relevance of sleep in maintaining overall health and managing chronic conditions by addressing hormonal imbalances, metabolic function, and lifestyle factors.

### 3.5. Social Support

Social support is crucial for managing CKM syndrome, obesity, CVD, diabetes, and CKD by promoting well-being and fostering healthy behaviors. The interconnected nature of CKM syndrome necessitates lifestyle interventions bolstered by robust social support networks [24]. SDOH, such as community engagement and interpersonal relationships, significantly influence individual health behaviors, which are essential for the management and prevention of CKM-related issues [4,18,20,37].

Social support plays a pivotal role in enhancing adherence to lifestyle modifications, offering motivation and accountability. Group-based interventions in weight management programs demonstrate superior outcomes by leveraging shared experiences and peer support, which promote physical activity, healthier dietary choices, and reduce psychological barriers within a supportive community environment. Moreover, addressing social determinants is critical for ensuring equitable access to resources, thereby improving the effectiveness of CKM management strategies [4,18,20,37]. By integrating social determinants into personalized intervention strategies, such approaches not only enhance health outcomes but also ensure that they are accessible to diverse populations. This emphasizes the importance of social support as a fundamental component in the successful management of CKM syndrome and related health conditions.

### 3.6. Avoidance of Risky Substances

Avoiding risky substances like tobacco, alcohol, and illicit drugs is crucial in mitigating detrimental effects on cardiovascular and kidney function, particularly within the scope of CKM syndrome management. Tobacco use exacerbates inflammation, oxidative stress, endothelial dysfunction, and atherosclerosis, significantly elevating CVD risk and accelerating CKD progression. Smoking is known to reduce glomerular filtration rates and increase proteinuria, complicating the management of CKM syndrome [4,131].

Excessive alcohol consumption is associated with an array of metabolic disruptions, such as hypertension, dyslipidemia, and insulin resistance, leading to obesity, diabetes, and subsequent cardiovascular disease. Alcohol’s toxic effects on the liver contribute to metabolic dysfunction, increased inflammation, and insulin resistance, further amplifying the risk of CKM syndrome. Conversely, moderating alcohol intake is beneficial, improving glycemic control and lipid profiles while reducing the burden on the cardiovascular and renal systems [2,4,38,135]. This comprehensive approach highlights the critical importance of lifestyle choices in CKM syndrome management, emphasizing the role of substance avoidance in improving health outcomes.

The reduction in risky substance use requires more than clinical advice; it demands robust policy support. The World Health Organization’s Framework Convention on Tobacco Control (FCTC) serves as the gold standard for such structural interventions. By coordinating demand-reduction measures, such as taxation and bans on advertising, with supply-side restrictions, the FCTC has successfully curbed tobacco prevalence globally. This policy framework offers a vital roadmap for addressing other CKM drivers [165]. Just as the FCTC targeted the tobacco industry’s commercial practices to reduce cardiovascular risk, similar regulatory approaches are likely needed to address the ‘industrial epidemic’ of ultra-processed foods and commercially driven sedentary behavior. Integrating these public health policies with clinical Lifestyle Medicine creates a ‘syndemic’ approach capable of addressing the root causes of CKM syndrome.

## 4. Conclusions

It is evident that CKM syndrome is complex syndrome characterized by a network of interconnected mechanisms, including but not limited to inflammation, oxidative stress, insulin resistance, and renin–angiotensin–aldosterone system dysregulation. As such, effective management requires a comprehensive, multifaceted approach that addresses these interconnected pathways [4,6,13,35,38]. This approach should integrate lifestyle medicine interventions not merely as a complement to pharmacological treatments for blood pressure, lipid regulation, glycemic control, and weight management, but potentially as a primary driver for inducing positive adaptations in CKM-related mechanisms [13,35,60,61,100]. In addition, this approach needs to consider the burden that the marginalized communities bear in terms of equitable and culturally tailored approaches [18,19,37]. Sex-specific factors are frequently overlooked in CKM-related research, with women—particularly post-menopause—receiving less attention due to hormonal fluctuations. Building on this need for personalization, our companion review synthesizes exercise dosing across CKM progression using a stage- and sex-informed lens [11]. This research gap in lipid metabolism and insulin sensitivity with respect to unique physiological and hormonal female contexts underscores the need for personalized genomic and nutraceutical lifestyle medicine interventions [13,14,17,166,167,168,169]. This personalization extends to physical activity, where sexual dimorphism in substrate metabolism and hormonal regulation dictates distinct responses to training intensity; for instance, men often exhibit greater visceral fat reduction from high-intensity intervals, while women may derive distinct vascular benefits from continuous moderate-intensity protocols [166,167,168]. Stage-specific exercise frameworks are increasingly incorporating sex-specific considerations relevant to CKM staging and training response [11].

While lifestyle interventions demonstrate promise, their clinical integration faces barriers, including prioritization of pharmacological treatments and inadequate protocol inclusion. Personalized strategies that align with individual preferences and leverage digital tools for adherence can overcome these challenges [49,73,111,170,171]. A holistic approach that incorporates lifestyle medicine and pharmacological advancements can optimize outcomes, reduce disparities, and improve quality of life in CKM populations [48,50,57,73,111].

Future research should investigate the integration of digital health innovations, for instance, electromyographic wearables that detect psychosocial interventions in real time [172], examine sex-specific responses to exercise, and assess the long-term effects of multimodal interventions. It should incorporate preregistered, adequately powered replication designs, building on the methodological roadmaps with strong methodological rigor and replicability to ensure the robustness of findings that underpin lifestyle medicine approaches [68,69,173,174,175]. By adopting a multidisciplinary framework, CKM management can become more effective and equitable, addressing the syndrome’s global burden comprehensively. Only when these interventions are integrated within a lifestyle medicine framework and complemented by pharmacological therapies may CKM’s root causes be adequately addressed [6,38,41,72,176].

## Figures and Tables

**Figure 1 healthcare-14-00051-f001:**
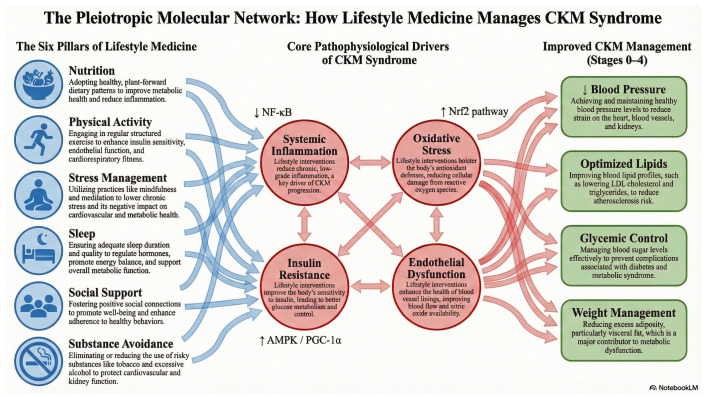
The Pleiotropic Molecular Network: How Lifestyle Medicine Manages CKM Syndrome. The diagram depicts a pleiotropic molecular network in which the six pillars of Lifestyle Medicine, nutrition, physical activity, stress management, sleep, social support, and avoidance of risky substances, act as inputs (**left**) that simultaneously modulate multiple upstream signaling pathways. These inputs influence key molecular targets, including AMP-activated protein kinase (AMPK)/peroxisome proliferator-activated receptor γ coactivator-1α (PGC-1α), nuclear factor erythroid 2-related factor 2 (Nrf2), endothelial nitric oxide synthase (eNOS), and nuclear factor κB (NF-κB), thereby attenuating systemic inflammation, oxidative stress, insulin resistance, and endothelial dysfunction (**center**). Together, these mechanisms converge to improve core CKM clinical phenotypes, blood pressure, lipid profiles, glycemic control, and weight management (**right**).

## Data Availability

All data generated or analyzed during this study are included in this published article.

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
