# Peer review of "Translational Lifestyle Medicine Approaches to Cardiovascular–Kidney–Metabolic Syndrome"

_healthcare, 2025, doi:10.3390/healthcare14010051_

Round 1

Reviewer 1 Report

Comments and Suggestions for Authors

This manuscript provides a comprehensive and well-referenced narrative review of the relationship between lifestyle medicine (LM) and the management of Cardiovascular-Kidney-Metabolic (CKM) syndrome. The review is thorough, well-structured, and draws on extensive literature. A major strength of the manuscript relates to the comprehensive synthesis of the current research. It integrates a large body of literature across cardiovascular, renal, metabolic and lifestyle domains. It appears that there is a good description of the six pillars of LM which is also backed up with recent references. It also appears that the manuscript gives a strong emphasis on translational relevance with highlights on barriers to adoption of LM in clinical settings, digital health apps and precision medicine and genomics. However, even though there is a description of many mechanisms the manuscript does not provide a unified diagram or framework explaining how LM interventions map onto CKM pathophysiology. Addition of a figure or a conceptual model showing CKM pathways, i.e. inflammation, oxidative stress, dyslipidemia, insulin resistance and how LM will modulate each pathway will strengthen a lot the manuscript.  A better description and assessment of the studies being cited (strength of evidence and study limitations) will add to the strength of the manuscript. Finally, it would be beneficial if a reduction in citation density where multiple similar sources are listed takes place.

Author Response

I thank the Editor and Reviewers for their thoughtful and constructive comments on our manuscript. I am grateful for the careful reading and the detailed suggestions, which have helped me significantly strengthen the mechanistic depth, visual clarity, and translational relevance of this work.

Below I respond point-by-point. Reviewer comments are reproduced in italics, followed by my responses. All changes mentioned below have been incorporated into the revised manuscript and are highlighted in yellow for ease of review.

Reviewer #1

Comment 1.1: This manuscript provides a comprehensive and well-referenced narrative review of the relationship between lifestyle medicine (LM) and the management of Cardiovascular-Kidney-Metabolic (CKM) syndrome. The review is thorough, well-structured, and draws on extensive literature. A major strength of the manuscript relates to the comprehensive synthesis of the current research. It integrates a large body of literature across cardiovascular, renal, metabolic and lifestyle domains. It appears that there is a good description of the six pillars of LM which is also backed up with recent references. It also appears that the manuscript gives a strong emphasis on translational relevance with highlights on barriers to adoption of LM in clinical settings, digital health apps and precision medicine and genomics. However, even though there is a description of many mechanisms the manuscript does not provide a unified diagram or framework explaining how LM interventions map onto CKM pathophysiology. Addition of a figure or a conceptual model showing CKM pathways, i.e. inflammation, oxidative stress, dyslipidemia, insulin resistance and how LM will modulate each pathway will strengthen a lot the manuscript.  

Response: I agree that a visual synthesis was necessary to anchor the text and illustrate the interconnectedness of these pathways.

  • Revision: I have added a new Figure 1, titled "The Pleiotropic Molecular Network: How Lifestyle Medicine Manages CKM Syndrome" This diagram explicitly maps the six pillars of Lifestyle Medicine to the core CKM pathophysiological targets (inflammation, oxidative stress, insulin resistance, and endothelial dysfunction) and demonstrates how they converge to improve clinical outcomes.
  • Location: Section 2.1 (Understanding CKM Syndrome), [Page 4, line 145].

Comment 1.2: A better description and assessment of the studies being cited (strength of evidence and study limitations) will add to the strength of the manuscript.

Response: Thank you for this suggestion. To provide a clearer assessment of the evidence quality, I have created a comprehensive summary table.

  • Revision: I added Table 1: Translational Lifestyle Medicine evidence relevant to CKM syndrome. This table summarizes key studies (including large-scale trials like PROPEL and Look AHEAD), details the target mechanisms, and explicitly includes a column for "Translational Insight / Limitation" to critically appraise the strength and applicability of the evidence.
  • Location: Section 2.2 (Translational Approaches), [Page 5–9].

Comment 1.3: Finally, it would be beneficial if a reduction in citation density where multiple similar sources are listed takes place.

Response: We have carefully audited the manuscript to remove "citation clusters" (lists of multiple redundant references). We have retained only the most relevant, recent, or seminal sources for each statement to improve readability.

  • Revision: Citation density has been reduced throughout, particularly in the Introduction and Section 3.2 (Physical Activity).

Reviewer 2 Report

Comments and Suggestions for Authors

Thank you to the authors for their efforts and contributions in this field. I have several suggestions that may help improve the manuscript:

  1. Lack of figures and tables
    Review articles typically include multiple figures and tables to help readers grasp key concepts and summarize complex information. However, this manuscript is presented entirely in textual form, which reduces readability and makes the review appear less formal. I strongly recommend adding schematic diagrams, summary tables, or graphical overviews to enhance the clarity of the article.

  2. Insufficient discussion of molecular mechanisms
    The manuscript provides limited explanation of the molecular and biochemical pathways through which exercise or nutrition exert their effects. To increase scientific depth, please expand this section and cite relevant literature (e.g., the study with DOI: 10.1016/j.jff.2023.105702) to better elucidate the underlying molecular biology mechanisms.

  3. Issues with citation format and reference selection
    Please review the citation format and the appropriateness of the cited evidence. For example, the clustered citation in line 372 (e.g., “[4, 5, 7, 47, 59, 63, 82, 114, 132, 143, 152]”) is overly lengthy and unnecessary. Only the most direct and authoritative sources should be cited. Additionally, Reference 4 is cited multiple times, which may not be justified. It is advisable to adopt a more selective and precise approach to citations.

  4. Limited novelty and depth
    The novelty of the manuscript appears limited. While it covers a broad range of topics, the depth of critical analysis is relatively weak, and some sections read as overly general. Although this does not preclude its publication as a review article, the authors may consider strengthening original insights and providing a more critical and comprehensive analysis to distinguish this work from generic or automatically generated summaries.

Author Response

I thank the Editor and Reviewers for their thoughtful and constructive comments on our manuscript. I am grateful for the careful reading and the detailed suggestions, which have helped me significantly strengthen the mechanistic depth, visual clarity, and translational relevance of this work.

Below I respond point-by-point. Reviewer comments are reproduced in italics, followed by my responses. All changes mentioned below have been incorporated into the revised manuscript and are highlighted in yellow for ease of review.

Reviewer #2

Comment 2.1: Lack of figures and tables. Review articles typically include multiple figures and tables to help readers grasp key concepts and summarize complex information. However, this manuscript is presented entirely in textual form, which reduces readability and makes the review appear less formal. I strongly recommend adding schematic diagrams, summary tables, or graphical overviews to enhance the clarity of the article.

Response: As noted in our response to Reviewer 1, I have significantly enhanced the visual presentation of the manuscript.

  • Revision: I added Figure 1 (Conceptual Framework) and Table 1 (Summary of Key Evidence) to break up the textual density and provide accessible visual summaries of the complex relationships discussed.
  • Location: Section 2.1 [Page 4, line 145] and Section 2.2 [Page 5–9].

Comment 2.2: Insufficient discussion of molecular mechanisms. The manuscript provides limited explanation of the molecular and biochemical

pathways through which exercise or nutrition exert their effects. To increase scientific depth, please expand this section and cite relevant literature (e.g., the study with DOI: 10.1016/j.jff.2023.105702) to better elucidate the underlying molecular biology mechanisms.

Response: I appreciate this specific guidance to deepen the scientific rigor of the review. I have expanded the discussion on molecular signaling pathways.

  • Revision: We added a new subsection, Section 2.4: Molecular Mechanisms of Lifestyle Interventions. This section details specific pathways, including PI3K/Akt/eNOS (vascular), AMPK/PGC-1α (metabolic), and Nrf2 (antioxidant).
  • Revision: We explicitly incorporated the suggested literature regarding "exercise-mimetics," discussing how specific nutrients (e.g., caffeine) induce irisin expression via the AMPK/PGC-1α pathway, citing Liu et al. (2023) as requested.
  • Location: Section 2.4, [Page 10, line 284-311].

Comment 2.3: Issues with citation format and reference selection. Please review the citation format and the appropriateness of the cited evidence. For example, the clustered citation in line 372 (e.g., “[4, 5, 7, 47, 59, 63, 82, 114, 132, 143, 152]”) is overly lengthy and unnecessary. Only the most direct and authoritative sources should be cited. Additionally, Reference 4 is cited multiple times, which may not be justified. It is advisable to adopt a more selective and precise approach to citations.

Response: I have addressed these formatting issues.

  • Revision: We have removed the clustered citations referenced (formerly line 372) and reduced the redundancy of frequently cited sources (such as Reference 4), prioritizing primary literature over general reviews where appropriate.

Comment 2.4: Limited novelty and depth. The novelty of the manuscript appears limited. While it covers a broad range of topics, the depth of critical analysis is relatively weak, and some sections read as overly general. Although this does not preclude its publication as a review article, the authors may consider strengthening original insights and providing a more critical and comprehensive analysis to distinguish this work from generic or automatically generated summaries.

Response: I have strengthened the critical analysis by moving beyond descriptive summaries to address systemic barriers and policy frameworks. The addition of the Commercial Determinants of Health (Section 3.1, page 12, line 374-385) and the Policy/FCTC discussion (Section 3.6, page 16, line 538-548) provides a novel, systems-level perspective on why LM often fails in isolation, distinguishing this work from generic clinical summaries.

Reviewer 3 Report

Comments and Suggestions for Authors

Dear author,

This manuscript make me ponder how useless seems to the LM medicine while treating patients. In theory, LM looks like a "magic bullet", but it rarely render is alleged benefits. LM is not a "new thing"! Besides cultural approaches, there is no mention of the role of industries that strongly hinders LM results due to profit!

For example, last month, The Lancet published a paper series where ultraprocessed food effects on health are exposed and several policies are suggested. In the series, the policies mainly target the industry producing them (url: https://www.thelancet.com/series-do/ultra-processed-food)! In addition, this reviewer is quite familiar with the FCTC (Framework Convention on Tobbaco Control). If you read it, you will realized that several of its articles target Tobacco Industry. The FCTC is very known regarding non communicable disease control.

Could you please give us a couple of words regarding this issue? Of course, with literature backing them!

Author Response

I thank the Editor and Reviewers for their thoughtful and constructive comments on our manuscript. I am grateful for the careful reading and the detailed suggestions, which have helped me significantly strengthen the mechanistic depth, visual clarity, and translational relevance of this work.

Below I respond point-by-point. Reviewer comments are reproduced in italics, followed by my responses. All changes mentioned below have been incorporated into the revised manuscript and are highlighted in yellow for ease of review.

Reviewer #3

Comment 3.1: Dear author,

This manuscript make me ponder how useless seems to the LM medicine while treating patients. In theory, LM looks like a "magic bullet", but it rarely render is alleged benefits. LM is not a "new thing"! Besides cultural approaches, there is no mention of the role of industries that strongly hinders LM results due to profit! For example, last month, The Lancet published a paper series where ultraprocessed food effects on health are exposed and several policies are suggested. In the series, the policies mainly target the industry producing them (url: https://www.thelancet.com/series-do/ultra-processed-food)!

Response: I sincerely appreciate this critical perspective. I agree that individual lifestyle choices do not exist in a vacuum and are heavily influenced by industrial drivers.

  • Revision: I added a robust discussion on the Commercial Determinants of Health in the Nutrition section. I explicitly cite the recent 2025 Lancet series on Ultra-Processed Foods (UPFs) to highlight how the industrial food environment disrupts satiety mechanisms and creates systemic barriers to adherence.
  • Excerpt: "Recent analyses, particularly the Lancet series on ultra-processed foods (UPFs), highlight that the global rise in CKM syndrome is driven not merely by a failure of individual willpower, but by an industrial food environment that prioritizes profit over metabolic health."
  • Location: Section 3.1 (Nutrition), [Page 12, line 374-385].

Comment 3.2: In addition, this reviewer is quite familiar with the FCTC (Framework Convention on Tobbaco Control). If you read it, you will realized that several of its articles target Tobacco Industry. The FCTC is very known regarding non communicable disease control. Could you please give us a couple of words regarding this issue? Of course, with literature backing them!

Response: Thank you for highlighting this crucial policy framework. I agree that the FCTC serves as the gold standard for the type of structural intervention required for CKM.

  • Revision: I added a new paragraph in the Avoidance of Risky Substances section discussing the WHO Framework Convention on Tobacco Control (FCTC). We suggest that this policy model—coordinating demand reduction with supply-side restrictions—offers a vital roadmap for addressing other CKM drivers, such as the "industrial epidemic" of ultra-processed foods.
  • Location: Section 3.6 (Avoidance of Risky Substances), [Page 16, line 538-548].

Round 2

Reviewer 2 Report

Comments and Suggestions for Authors

ok